# GENERATING TRANSFERABLE AND STEALTHY ADVERSARIAL PATCH VIA ATTENTION-GUIDED ADVERSARIAL INPAINTING

## ABSTRACT

Adversarial patch attacks can fool the face recognition (FR) models via small patches. However, previous adversarial patch attacks often result in unnatural patterns that are easily noticeable. Generating transferable and stealthy adversarial patches that can efficiently deceive the black-box FR models while having good camouflage is challenging because of the huge stylistic difference between the source and target images. To generate transferable, natural-looking, and stealthy adversarial patches, we propose an innovative two-stage attack called *Adv-Inpainting*, which extracts style features and identity features from the attacker and target faces, respectively and then fills the patches with misleading and inconspicuous content guided by attention maps. In the first stage, we extract multi-scale style embeddings by a pyramid-like network and identity embeddings by a pretrained FR model and propose a novel Attention-guided Adaptive Instance Normalization layer (AAIN) to merge them via background-patch cross-attention maps. The proposed layer can adaptively fuse identity and style embeddings by fully exploiting priority contextual information. In the second stage, we design an Adversarial Patch Refinement Network (APR-Net) with a novel boundary variance loss, a spatial discounted reconstruction loss, and a perceptual loss to boost the stealthiness further. Experiments demonstrate that our attack can generate adversarial patches with improved visual quality, better stealthiness, and stronger transferability than state-of-the-art adversarial patch attacks and semantic attacks.

## 1 INTRODUCTION

Deep neural networks are widely used in many applications, such as image classification and face recognition (FR). Previous research has shown that they are vulnerable to adversarial examples that are created by adding elaborate noises, which are invisible to human eyes but can deceive the neural network (Madry et al., 2017; Carlini & Wagner, 2017). However, such attacks are impractical in real-world implementation because pixel-wise manipulation is usually infeasible. To attack real-world applications like face recognition systems, most researchers use adversarial patches rather than pixel-wise noises, which limits adversarial noises to a small area to be printable and can be physically used to attack FR systems (Komkov & Petiushko, 2021; Yang et al., 2020).

Stealthiness and transferability to black-box models are two crucial requirements for practical adversarial patch attacks. Although some methods have been proposed to enhance the transferability of adversarial patches (Wang et al., 2021; Dong et al., 2018; Zhao et al., 2021; Ma et al., 2023), they often produce unnatural styles and noticeable boundaries, making the patches easily detectable (see Figure 1(c-g)). Xiao et al. (2021) found that sampling from the latent space of generative models can enhance transferability. However, their method also did not take the camouflage into consideration, making the generated patches conspicuous (see Figure 1(f)).

Is there an inherent contradiction between stealthiness and transferability? Recently, researchers have found that editing deep features can boost transferability (Inkawhich et al., 2019; Jia et al., 2022). Based on these findings, we propose that the paradox between stealthiness and transferability can be effectively addressed by deep feature manipulation. However, because of the huge property difference between the source and target image, like age, gender, skin color, and expressions, blend-

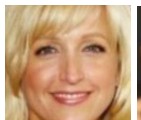 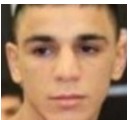 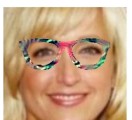 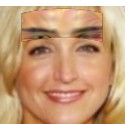 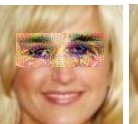 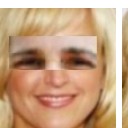 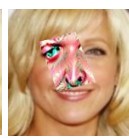 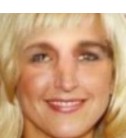

(a) Attacker  (b) Target  (c) Adv-Glasses  (d) Adv-Hat  (e) TIDIM  (f) GenAP  (g) SMAP  (h) **Ours**

Figure 1: Illustration of different adversarial patch attacks, including Adv-Glasses (Sharif et al., 2016), Adv-Hat (Komkov & Petiushko, 2021), TIDIM (Xie et al., 2019b)), GenAP (Xiao et al., 2021)), SMAP (Ma et al., 2023) and ours. The adversarial patches generated by Adv-Inpainting have higher transferability and stealthiness than previous state-of-the-art adversarial patch attacks.

ing the stylistic difference seamlessly while deceiving the FR model through a small patch remains a significant challenge. To improve attack transferability and patch stealthiness simultaneously, we propose a novel attack called *Adv-Inpainting* by employing two-stage attention-guided deep feature manipulation. Our method merges the deep features of the original and target face images while considering the context through an attention-guided adaptive instance normalization (AAIN) layer, which can fuse the identity features of the target image into the source image while maintaining the features not related to the identity, such as expressions and postures. Furthermore, compared with previous feature space attacks (Jia et al., 2022), the stealthiness of our attack is measured by the perceptual similarity (Zhang et al., 2018), which is more consistent with human perception and can be optimized simultaneously with adversary loss, making the training process more stable.

Adv-Inpainting consists of two stages. The first stage contains an attention-guided StyleGAN-based generative model (Att-StyleGAN) with a new AAIN layer to generate reasonable and transferable adversarial patches. To further improve the camouflage of the adversarial patches, we append a second stage named Adversarial Patch Refinement Network (APR-Net) to blend the adversarial patch with the source image seamlessly while maintaining the adversarial attributes. An example of the generated patch is shown in Figure 1(h). In summary, our contributions are as follows:

- We are the first to realize a transferable and stealthy adversarial patch attack. Our research indicates that filling reasonable content into the masked region through attention-guided deep feature manipulation can significantly enhance the transferability of the adversarial patches. Compared with previous attack methods, Adv-Inpainting is superior in generating photorealistic, stealthy, and highly transferable adversarial patches.

- We design a novel two-stage coarse-to-fine attack framework. The first stage employs a novel AAIN layer, which has a new mechanism to combine identity and style features. It can fully exploit prior information of the source image to generate transferable and stylistically consistent adversarial patches. In the second stage, dual spatial attention layers are utilized to fine-tune the adversarial patch to improve concealment further.

- Our research pioneers the adoption of perception distance to enhance the stealthiness of the generated patches and stabilize the training process of adversarial attacks. Additionally, we introduce a novel boundary variance loss that facilitates the fine-tuning of the patch, ensuring its coherency with the background.

- Experiments on various FR models demonstrate our method's effectiveness. Our approach significantly improves transferability and stealthiness and surpasses the state-of-the-art attack by a large margin. Moreover, our trained model can directly generate adversarial patches end-to-end, which is more efficient than previous generative attacks.

## 2 RELATED WORK

**Transferable Adversarial Patch Attack.** Different from the traditional strategy of finding adversarial examples, adversarial patch attacks confine the perturbation in a small area, enabling them to be printable and affixed onto the original object to deceive the classification model. For example, Sharif et al. (2016) employed meticulously crafted eyeglass frames to attack the face recognition model. Komkov & Petiushko (2021) confined the perturbation in a hat area and initialized the adversarial patch to the target face. However, these early attacks focus on white-box settings and have

low transferability to black-box models. Xie et al. (2019b) proposed TIDIM attack, which combines FGSM (Goodfellow et al., 2014) with random transformations to improve transferability. However, TIDIM did not consider the stealthiness of the patches. Xiao et al. (2021) shows that regularizing the adversarial patches on the latent space of StyleGAN can improve the transferability. However, GenAP did not consider the contextual information of the image either, resulting in conspicuous patches with evident stylistic differences from the original image. Furthermore, iterative sampling from the latent space of a face generator is very time-consuming.

**Generative Adversarial Attack.** Our attack is also related to generative adversarial attacks. To enhance the transferability of adversarial examples, some attacks perturb the high-level semantics using generative models, such as SemanticAdv (Qiu et al., 2020), Adv-Makeup (Yin et al., 2021) and Adv-attribute (Jia et al., 2022). While these attacks have demonstrated improved visual quality, their computational demands are high. For example, Adv-Makeup needs to generate a makeup dataset from a makeup generation model and train the attack model on multiple face recognition models. Adv-attribute needs to select the most important attributes and define the stealthiness loss and adversary loss in conflicting ways, making the training process unstable and need to adjust weights carefully during the training. Furthermore, SemanticAdv and Adv-attribute are difficult to realize in the physical world as accurately reproducing attributes like hairstyle and facial expression proves difficult.

**Image Inpainting and Image Composition.** Our work draws inspiration from state-of-the-art image inpainting and composition technologies. Yu et al. (2018) first proposed a contextual attention layer to generate consistent patches. Building upon this, Zeng et al. (2019) combined the attention layer and pyramid-context encoder to get the masked image's embedding. Xie et al. (2019a) presented learnable bidirectional attention maps to produce visually plausible inpainting content. Zhou et al. (2020) enhance the patch quality further through a dual spatial attention module, leveraging background-foreground cross attention and foreground-self attention. Additionally, our work also draws inspiration from image composition, particularly the module proposed by Ling et al. (2021) that explicitly captures the visual style from the background and applies it to the foreground. However, our work addresses a more challenging scenario than previous inpainting and composition tasks. While filling the masked region with coherent content remains important, we aim to ensure that the resulting inpainted image is correctly classified as the desired target label. This requirement is crucial for effective adversarial patch attacks in face recognition systems. Due to the great difference in age, gender, posture, skin color, and other characteristics between the source and the target image, simultaneously bridging the gap between the source and target image and fooling the face recognition model with a small patch is very challenging.

## 3 METHODOLOGY

We propose a two-stage attack called Adv-Inpainting to generate transferable and stealthy patches. In the first stage, we propose a novel framework consisting of a style encoder, an identity encoder, and an attention-guided StyleGAN (Att-StyleGAN) to generate transferable and natural patches. In the second stage, we propose an Adversarial Patch Refinement Network (APR-Net) to fine-tune the adversarial patch to improve its camouflage further while maintaining adversarial attributes.

### 3.1 ATTENTION-GUIDED STYLEGAN FOR ADVERSARIAL PATCH GENERATION

StyleGAN (Karras et al., 2019) has shown excellent ability to generate photorealistic images from noises. Recently, Xiao et al. (2021) utilized it to generate transferable adversarial patches. However, because StyleGAN is not designed for adversarial attacks, direct sampling noises from the latent space of the StyleGAN will generate unnatural and conspicuous patches (Figure 1(f)). To solve this problem, we propose a novel end-to-end adversarial patch generation framework. We will first introduce the architecture of the proposed framework and then introduce the design of loss functions.

**The architecture of Att-StyleGAN** As shown in Figure 2(a), Att-StyleGAN consists of a style encoder to extract style-related features, a pretrained identity encoder to extract identity features, and an attention-guided StyleGAN generator to generate transferable and photorealistic images. The style encode utilizes the the state-of-the-art style embedding framework, called pixel-to-Style-pixel

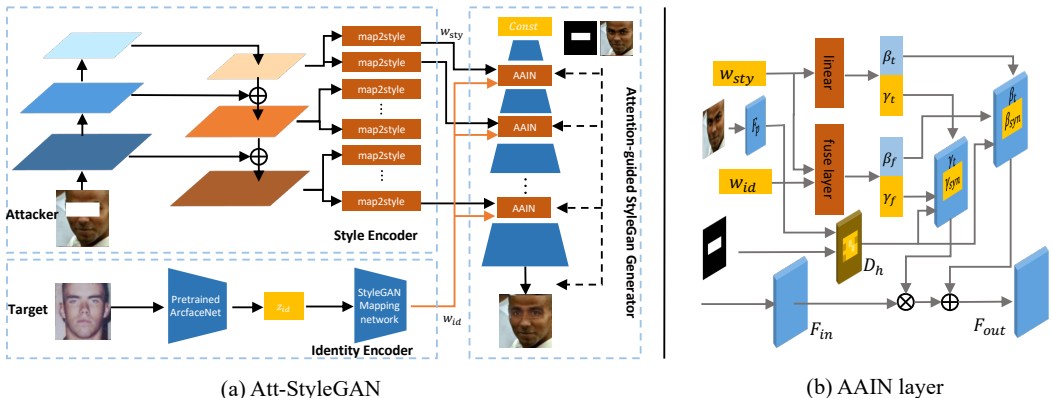

(a) Att-StyleGAN             (b) AAIN layer

Figure 2: (a) Overview of the first stage of Adv-Inpainting, called Att-StyleGAN, which contains a pSp encoder, an identity encoder, and an attention-guided StyleGAN generator. (b) Illustration of the proposed AAIN layer, which fuses the style embedding and identity embedding through a contextual attention map to generate transferable and stealthy patches.

(pSp) (Richardson et al., 2021) network, which adopts pyramid-like architectures to extract multi-scale features. It was initially designed for style transfer tasks. In this paper, we use it to extract features related to the posture and stylistic information, then map the features to style embeddings $w_{sty}$ in the $\mathcal{W}+$ space (Abdal et al., 2019) of the StyleGAN. Also, we use a pretrained face recognition model, e.g., Arcface (Deng et al., 2019), to extract the identity feature $z_{id}$ from the target image. We then map $z_{id}$ to $w_{id}$ in the $\mathcal{W}+$ space through a mapping network $G_m$. At last, style embeddings $w_{sty}$ and identity embeddings $w_{id}$ are fed into the AAIN layers. Let $x_{syn}$ be the synthesized image of the generator, $x_s$ be the source image, and $M$ be the mask, then the final output is

$$x_{out} = x_{syn} \cdot (1 - M) + x_s \cdot M. \tag{1}$$

**Attention-guided Adaptive Instance Normorlization** Adaptive instance normalization (AdaIN) (Huang & Belongie, 2017) can realize real-time style transfer and is widely used in image-to-image translation. The state-of-the-art face generative model, StyleGAN adopts AdaIN to generate high-fidelity images:

$$AdaIN(F_i, w) = \gamma_w \left( \frac{F_i - \mu(F_i)}{\sigma(F_i)} \right) + \mu_w, \tag{2}$$

where $F_i$ is the $i^{th}$ block's output of StyleGAN and is normalized over its spatial locations. $\gamma_w$ and $\mu_w$ are affine transformations of latent vector $w$ to stylize the feature map $F_i$. However, AdaIN is unsuitable for our attack because the context information of the source image is not considered.

To merge the identity embeddings of the target image and the style embeddings of the source image, such as expression and posture features, we propose a new attention-guided adaptive instance normalization layer (AAIN). The architecture of AAIN is shown in Figure 2(b). We first use two linear layers to get the attacker-related style vectors $\beta_t, \gamma_t$ and fused style vectors $\beta_f, \gamma_f$ from the style embedding $w_{sty}$ and the identity embedding $w_{id}$,

$$[\beta_t, \gamma_t] = Lin_{tex}(w_{sty}), \quad [\beta_f, \gamma_f] = Lin_{fuse}([w_{sty}, w_{id}]), \tag{3}$$

Although the fuse layer $Lin_{fuse}$ can mix the feature embeddings of the source and target images, it does not consider the contextual coherence with the background. To improve the coherence and camouflage of the adversarial patch, we compute the background-patch cross-attention map to fuse the styles further. As shown in Figure 2(b), We use the feature map $F_p$ of the source image, the normalized feature map $F_{in}$ from the previous block and the mask $M$ as the prior information to get the background-patch cross-attention map $D_h$:

$$D_h = \sigma(Conv([F^p, F^{in}]) \cdot (1 - M) + M, \tag{4}$$

where the hole of $M$ is set to 0, and the background of $M$ is set to 1. The $Conv$ is a convolution layer, and $\sigma$ is a sigmoid activation function. The synthesized style vectors for the patch area are :

$$\gamma_{syn} = D_h \gamma_t + (1 - D_h)\gamma_f, \quad \beta_{syn} = D_h \beta_t + (1 - D_h)\beta_f. \tag{5}$$

The total styles to stylize the normalized feature map $F_{in}$ are:

$$\gamma_i = M\gamma_t + (1 - M)\gamma_{syn}, \quad \beta_i = M\gamma_t + (1 - M)\gamma_{syn}. \tag{6}$$

Finally, the output of AAIN layer can be expressed as:

$$AAIN(F_i, F_p, w_{sty}, w_{id}) = \gamma_i(\frac{F_i - \mu(F_i)}{\sigma(F_i)}) + \mu_i. \tag{7}$$

**The loss functions of Att-StyleGAN**  In our attack, the goal is to make the source image with the adversarial patch be identified as the target image $x_t$. Therefore, we use the cosine similarity loss as the adversarial loss:

$$\mathcal{L}_{adv} = 1 - cossim(f(x_{out}), f(x_t)), \tag{8}$$

where $f(\cdot)$ is the white-box substitute model to be attacked.

We also use the semi-supervised learning strategy to train the Att-StyleGAN. Let $id(\cdot)$ be the identity function to map the face image to its labels. The recovery loss is expressed as:

$$\mathcal{L}_{rec} = \begin{cases} \frac{1}{2}\|x_{syn} - x_s\|_2^2, & if\ id(x_s) = id(x_t) \\ \frac{1}{2}\|x_{syn} \cdot M_d - x_s \cdot M_d\|_2^2, & if\ id(x_s) \neq id(x_t) \end{cases}, \tag{9}$$

where $M_d$ is a discounted mask, where the value outside the patch area is 1, and the value in the patch area is set to $\frac{1}{\alpha \cdot e^l}$, where $l$ is the distance from the pixel to the patch boundary, and $\alpha$ is set to 0.15 in our experiments. We use $M_d$ to force the pixel values near the patch boundary to be more consistent with the source image while giving more freedom to the inner pixels.

Furthermore, we include the Learned Perceptual Image Patch Similarity (LPIPS) loss (Zhang et al., 2018) to improve perceptual similarity between $x_{syn}$ and $x_s$, which computes the cosine distance of deep features via a pretrained VGG-Net. LPIPS loss can make the adversarial patches' quality and style more similar to the original content and is consistent with human perception, which enables a further blend between the source image and the patches to boost stealthiness:

$$\mathcal{L}_{\text{LPIPS}} = \Sigma_l \frac{1}{H_l W_l} \Sigma_{h,w} \|w_l \odot (\hat{y}_{hw}^l - \hat{y}_0^l hw)\|_2^2, \tag{10}$$

where $\hat{y}^l, \hat{y}_0^l$ are features of $x_{syn}$ and $x_s$ extracted from the pretrained VGG-Net's layer $l$. Finally, we include the discriminator loss $\mathcal{L}_{dis}$, which is the same as the StyleGAN. The final loss is:

$$L_{total} = \lambda_{adv}\mathcal{L}_{adv} + \lambda_{rec}\mathcal{L}_{rec} + \lambda_{\text{LPIPS}}\mathcal{L}_{\text{LPIPS}} + \lambda_{dis}\mathcal{L}_{dis}, \tag{11}$$

where $\lambda_{adv}, \lambda_{rec}, \lambda_{\text{LPIPS}}$ and $\lambda_{dis}$ are balance weights. During the training process, the pretrained FR model is fixed, and the style encoder, the mapping network $G_m$ and the generator are updated.

### 3.2 Adversarial Patch Refinement Network

In this paper, we propose an adversarial patch refinement network (APR-Net) to refine the generated adversarial patch further to make it more stealthy and coherent with the source image while maintaining the adversarial properties. As shown in Figure 3, APR-Net consists of a U-Net-like generator with dual spatial attention layers (Zhou et al., 2020) and a PatchGAN (Isola et al., 2017) discriminator. Dual spatial attention can greatly improve the naturalness of image inpainting through background-foreground cross attention and foreground-self attention. PatchGAN discriminator tries to classify if each patch with the same size in the reconstructed image is real or fake. Therefore, it can identify unnatural patches effectively and promote generated image qualities.

The adversarial loss $\mathcal{L}_{adv}$ is also included to maintain the adversarial properties, which is the same as the Eq. 8. The spatial discounted reconstruction loss utilizes a semi-supervised learning strategy to train the generator to recover the input image,

$$\mathcal{L}_{rec} = \begin{cases} \frac{1}{2}\|x_{refine} - x_s\|_2^2, & if\ id(x_s) = id(x_t) \\ \frac{1}{2}\|(x_{refine} - x_{out}) \cdot M_d\|_2^2, & if\ id(x_s) \neq id(x_t) \end{cases}, \tag{12}$$

where $x_{out}$ is the first stage's output, and $x_{refine}$ is the refined image. $M_d$ is a discounted mask. $\mathcal{L}_{\text{LPIPS}}$ is also used to improve the patch perceptual similarity with the source image.

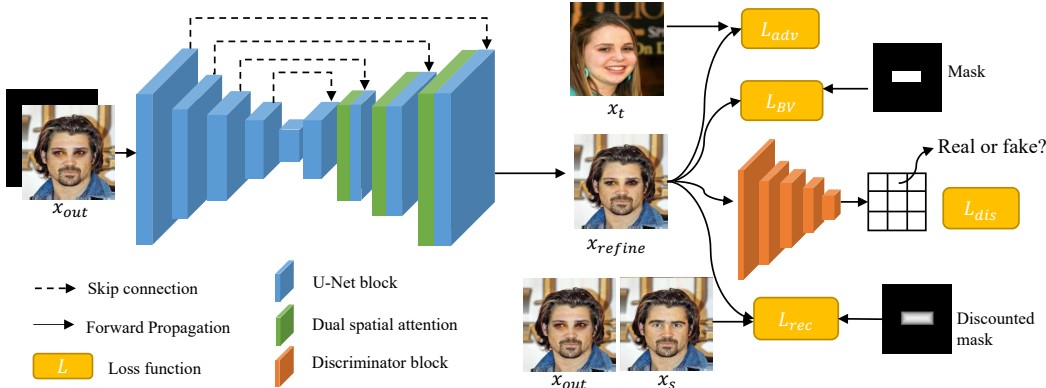

Figure 3: The architecture and loss functions of APR-Net. The $x_s$ is the source image. The $x_{out}$ is the output image of the first stage (see Eq. 1). The $x_{refine}$ is the output image of APR-Net. The adversarial patch in $x_{out}$ shows a subtle boundary and a little unnaturalness; in contrast, the adversarial patch in $x_{refine}$ is stylistically consistent and naturalistic.

**Boundary Variance Loss**   To encourage the APR-Net to refine adversarial patches to be seamlessly consistent with the background, we propose a new loss function called boundary variance loss, which computes the variance inside and outside the patch boundaries:

$$\mathcal{L}_{BV} = \frac{1}{2}(\|x_{l,t:b}-x_{l-1,t:b}\|_2^2+\|x_{r,t:b}-x_{r+1,t:b}\|_2^2+\|x_{l:r,t}-x_{l:r,t-1}\|_2^2+\|x_{l:r,b}-x_{l:r,b+1}\|_2^2), \quad (13)$$

where $x = x_s \cdot M + x_{refine} \cdot (1 - M)$, and $l, r, t, b$ are the left, right, top, and bottom boundary of the mask. Finally, we include the PatchGAN discriminator loss $\mathcal{L}_{dis}$ with a gradient penalty term (Gulrajani et al., 2017) to distinguish each square patch of the refined image as real patches, as shown in Figure 3. The total loss function of APR-Net is:

$$L_{total} = \lambda_{adv}\mathcal{L}_{adv} + \lambda_{rec}\mathcal{L}_{rec} + \lambda_{BV}\mathcal{L}_{BV} + \lambda_{dis}\mathcal{L}_{dis} + \lambda_{\text{LPIPS}}\mathcal{L}_{\text{LPIPS}}. \quad (14)$$

## 4 EXPERIMENTS

This section introduces the experimental setup, including the dataset used, the victim models being attacked, and the compared attacks. We then present the quantitative results of black-box transferability and stealthiness. Additionally, we show the results of ablation studies on different network modules and loss functions. Lastly, we present the attack results on defense models.

### 4.1 EXPERIMENTAL SETUP

**Datasets**   We chose three datasets with different resolutions to evaluate the effectiveness of our attack.(1) LFW dataset (Huang et al., 2007), which contains around 13,000 images. The image resolution is 128×128. (2) CelebA dataset (Liu et al., 2015) with more than 20,000 images. The image resolution is 256×256. (3) FFHQ dataset (Karras et al., 2019) with over 70,000 high-quality images at 1024×1024 resolution.

**Evaluation metric**   We use the attack success rate (ASR) and the perceptual similarity of the source images with and without the adversarial patch as the evaluation metric. To evaluate the attack transferability, we first randomly generate $N$ source-target pairs such that $id(x_s) \neq id(x_t)$ and then compute ASR through:

$$ASR = \frac{\Sigma_i^N \mathbb{1}(cossim(f(x_t), f(x_{refined})) > \tau))}{N} \times 100\%, \quad (15)$$

where $\mathbb{1}(\cdot)$ is an indicator function. The $cossim$ measures the cosine similarity between two identity vectors. The value of $\tau$ is set as the threshold that can acquire the highest accuracy on the classified dataset through $K$-fold cross-validation, *i.e.* ArcFace (0.23), CosFace (0.26), MobileFace(0.19),

| Dataset | LFW Dastaset | | | | CelebA Dataset | | | | FFHQ Dataset | | | |
|---|---|---|---|---|---|---|---|---|---|---|---|---|
| Victim Model | ArcFace | CosFace | MobFace | FaceNet | ArcFace | CosFace | MobFace | FaceNet | ArcFace | CosFace | MobFace | FaceNet |
| FGSM | 87.55 | 8.26 | 3.75 | 10.92 | 89.45 | 4.56 | 6.14 | 8.44 | 92.43 | 8.64 | 5.60 | 10.32 |
| PGD | 98.73 | 13.45 | 11.80 | 12.60 | 94.56 | 12.57 | 9.90 | 13.50 | 95.45 | 10.64 | 8.85 | 12.45 |
| C&W | 97.65 | 15.53 | 11.92 | 13.45 | 95.67 | 14.56 | 11.74 | 14.62 | 89.65 | 12.55 | 10.60 | 16.48 |
| MI-FGSM | 95.48 | 16.55 | 16.27 | 14.52 | 98.90 | 15.89 | 8.23 | 15.30 | 98.20 | 15.35 | 10.24 | 13.25 |
| Adv-Glasses | 47.43 | 3.55 | 4.60 | 8.20 | 45.43 | 2.41 | 5.67 | 9.10 | 43.43 | 13.23 | 14.60 | 12.20 |
| Adv-Hat | 76.46 | 9.40 | 3.10 | 4.80 | 75.23 | 14.54 | 8.56 | 4.74 | 56.43 | 23.40 | 22.10 | 14.21 |
| TIDIM | 89.68 | 25.48 | 24.88 | 32.56 | 88.34 | 27.86 | 21.77 | 22.53 | 85.38 | 25.40 | 24.60 | 32.52 |
| Gen-AP | 75.65 | 22.56 | 19.90 | 8.20 | 82.45 | 21.34 | 24.40 | 25.82 | 81.75 | 32.56 | 29.90 | 12.20 |
| SMAP | 89.33 | 29.89 | 23.56 | 27.64 | 88.56 | 30.35 | 27.88 | 29.57 | 92.45 | 34.67 | 26.75 | 28.32 |
| SemanticAdv | 88.54 | 8.90 | 13.92 | 5.87 | 88.5 | 22.45 | 19.42 | 9.02 | 84.34 | 18.21 | 13.19 | 7.88 |
| Adv-Makeup | 85.79 | 13.47 | 19.01 | 8.80 | 85.73 | 21.60 | 14.60 | 9.50 | 86.70 | 23.41 | 16.01 | 10.70 |
| Adv-Attribute | 86.36 | 28.70 | 26.50 | 25.90 | 86.36 | 23.52 | 25.65 | 31.80 | 84.45 | 28.72 | 22.52 | 31.45 |
| Ours w.o. AAIN | 89.57 | 29.78 | 28.12 | 34.58 | 83.67 | 29.46 | 24.15 | 36.17 | 88.50 | 25.68 | 26.21 | 35.60 |
| Ours w.o. APT | 90.55 | 35.71 | 30.82 | 37.41 | 84.56 | 34.45 | 33.05 | 38.20 | 95.52 | 37.46 | 33.42 | 40.60 |
| Adv-Inpainting | 92.34 | **40.64** | **35.50** | **42.95** | 93.40 | **37.10** | **38.36** | **42.30** | 95.45 | **37.97** | **35.45** | **42.40** |

Table 1: The attack success rates (%) of impersonation attacks for the different face recognition models on the LFW and CelebA datasets. We chose the ArcFace model as the white-box model to train our models and evaluate the transferability of the other three FR models. We compare our attack with gradient-based, patch-based, and semantic-based attacks respectively.

and FaceNet(0.36) on the CelebA dataset. The perceptual distances are more suitable for measuring semantic perturbations than the Mean Squared Error (MSE) distance, which is more consistent with human perception. LPIPS (Zhang et al., 2018) outperforms previous perceptual metrics by a large margin. Therefore, we use LPIPS to measure the perceptual similarity between the source and the adversarial image. We also calculate the Fréchet Inception Distance (FID)(Chong & Forsyth, 2020), which is widely used to assess the quality of images created by generative models.

**Compared attacks** We chose three different kinds of adversarial attacks as the competitors: gradient-based, patch-based, and semantic-based attacks. The gradient-based attacks include FGSM (Goodfellow et al., 2014), PGD (Madry et al., 2017), C&W (Carlini & Wagner, 2017) and MI-FGSM (Dong et al., 2018). The patch-based attacks include Adv-Glass (Sharif et al., 2016), Adv-Hat (Komkov & Petiushko, 2021), TIDIM (Xie et al., 2019b) and Gen-AP (Xiao et al., 2021) and SMAP (Ma et al., 2023). The semantic-based attacks include Adv-Makeup (Yin et al., 2021), Semantic-Adv (Qiu et al., 2020) and Adv-Attribute (Jia et al., 2022). For the gradient-based attacks, we set the maximal perturbations as $\epsilon = 0.2$, and for the patch attacks, we set the maximum patch size as $50 \times 100$ for $256 \times 256$ images.

**Implementation details** Our framework is written in PyTorch and is trained on NVIDIA 3090 Ti. We compared different optimizers, such as SGD, Adam, and AdamW, and finally chose AdamW as the optimizer. For the optimizer, we set the initial learning rate as 0.001, the $\beta_1$ as 0.9, and the $\beta_2$ as 0.999. For the weights of loss functions, the $\lambda_{adv}, \lambda_{rec}, \lambda_{\text{LPIPS}}, \lambda_{dis}$ are set to 1 and $\lambda_{BV}$ is set to 0.01. For different datasets with varied resolutions, the Att-StyleGAN are trained respectively. For example, the CelebA dataset's resolution is $256 \times 256$. Therefore, the Att-StyleGAN generator has seven blocks. Meanwhile, the dimension of $w_{id}$ and $w_{sty}$ are both $14 \times 512$, and the maximum mask size is set as 50*100 to be the same as the setting in GenAP (Xiao et al., 2021). We choose four representative face recognition models, such as Facenet(Schroff et al., 2015), ArcFace (Deng et al., 2019), CosFace (Wang et al., 2018) and MobileFace (Chen et al., 2018), as the victim models. We implement all the FR models through the official codes or utilize the pretrained models.

## 4.2 EXPERIMENT RESULTS

**ASR and transferability** We compare our attack with the gradient-based, patch-based, and semantic-based attacks respectively. The ASR results are shown in Table 1. We use the ArcFace model as the white-box model. Then, we evaluate the black-box attack success rate on the other three FR models, including CosFace, MobileFace, and FaceNet. The results show that the transferability of our attack outperforms the previous state-of-the-art adversarial patch attacks, GenAP and SMAP, by large margins. Our attack also significantly outperforms the state-of-the-art semantic at-

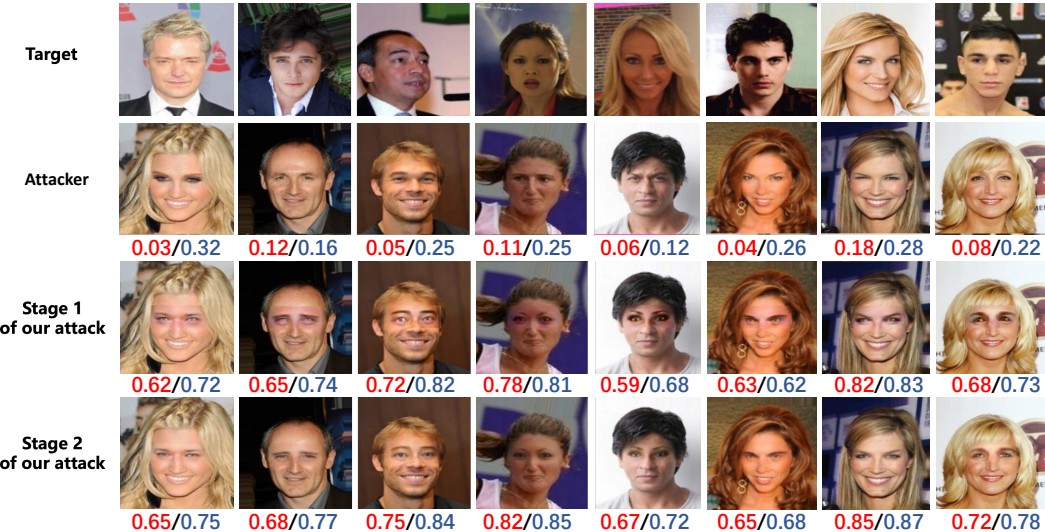

Figure 4: Visualizations of Adv-Inpainting. The below numbers are the identity features' cosine similarity with the target images. The red numbers are on the white-box model ArcFace, and the blue numbers are on the black-box model FaceNet. More visualization is shown in the appendix.

| Dataset | LFW | | | CelebA | | | FFHQ | | |
|---|---|---|---|---|---|---|---|---|---|
| Metric | MSE(↓) | FID(↓) | LPIPS(↓) | MSE(↓) | FID(↓) | LPIPS(↓) | MSE(↓) | FID(↓) | LPIPS(↓) |
| Adv-Glasses | 83.40 | 106.70 | 74.90 | 78.20 | 104.37 | 88.47 | 70.45 | 105.10 | 78.34 |
| Adv-Hat | 86.87 | 123.40 | 83.10 | 84.80 | 125.32 | 86.75 | 89.56 | 124.74 | 98.67 |
| TIDIM | 89.60 | 134.00 | 84.56 | 87.50 | 136.30 | 87.86 | 90.45 | 132.53 | 99.56 |
| Gen-AP | 75.65 | 92.56 | 88.90 | 88.20 | 99.45 | 91.34 | 85.80 | 94.40 | 98.90 |
| SMAP | 89.15 | 127.34 | 89.25 | 84.64 | 125.36 | 84.52 | 91.24 | 128.45 | 92.42 |
| Adv-Makeup | 81.69 | 93.40 | 81.10 | 80.80 | 105.7 | 89.37 | 81.60 | 102.50 | 81.60 |
| SemanticAdv | 97.54 | 93.53 | 78.90 | 83.57 | 107.52 | 93.56 | 89.02 | 118.42 | 95.64 |
| Adv-Attribute | 80.36 | 91.97 | 75.34 | 81.67 | 94.89 | 88.46 | 79.37 | 94.86 | 85.45 |
| Ours w.o. AAIN | 76.57 | 77.56 | 65.45 | 74.32 | 74.34 | 66.45 | 73.45 | 78.30 | 67.31 |
| Ours w.o. APT | 78.55 | 85.34 | 72.70 | 82.33 | 86.56 | 73.49 | 82.28 | 86.66 | 75.76 |
| Adv-Inpainting | **63.46** | **72.76** | **58.35** | **61.29** | **70.54** | **57.89** | **57.20** | **71.46** | **62.26** |

Table 2: Evaluation of the adversarial patch stealthiness. We compute the MSE ($l_2$), FID and LPIPS distances between the source and adversarial images for different adversarial patch and semantic attacks. Compared with the state-of-the-art adversarial patch attack Gen-AP, our attack significantly promotes the visual quality of the adversarial patches.

tack, Adv-Attribute. Moreover, our attack has a high transferability on the FaceNet, which previous attacks did not perform very well. We also use CosFace as the white-box model and leave the results in the appendix.

**Stealthiness Evaluation** As shown in Figure 4, compared with the previous state-of-the-art adversarial patch attacks, TIDIM and Gen-AP, our attack is more stealthy because we modify the deep embedding rather than the image directly and, therefore, doesn't have unnatural patterns and is more stylistically consistent with the background. Our attack can successfully generate stealthy and coherent patches for faces of different genders, postures, expressions, and ages.

Moreover, we believe that natural images are less likely to attract human attention and, thus, more stealthy. Therefore, we use the MSE, FID (Heusel et al., 2017) and LPIPS (Zhang et al., 2018) as the stealthiness evaluation metrics (Table 2). The results show that our attack outperforms the state-of-the-art transferable adversarial patch attacks by large margins. Moreover, our attack also outperforms previous transferable semantic attacks, SemanticAdv and Adv-Attribute, which we think is because we perturb the image through a small patch rather than modifying the whole image.

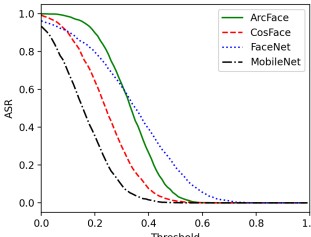 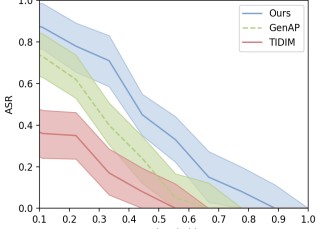 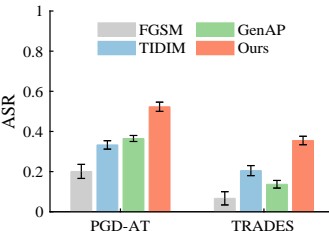

Figure 5: ASR-threshold curve on different models.

Figure 6: ASR-threshold curve on different attacks.

Figure 7: Black-box ASRs on robust models.

**Ablation Study** We investigate the effect of the AAIN and APR-Net on transferability and stealthiness, respectively. The third-to-last lines in Table 1 and 2 indicate the ASR and stealthiness without the AAIN, showing that AAIN can greatly improve the black-box ASR. The penultimate lines in Table 1 and 2 show the ASR and stealthiness without APR-Net, illustrating that APR-Net effectively enhances naturalness without harming the ASR. Figure 5 presents the ASR-threshold curve for different FR models to illustrate their robustness against our attack. MobileFace's ASR drops quickly when $\tau$ increases, indicating its higher robustness to the attack, while FaceNet is the most vulnerable model to our attack. Figure 6 shows the ASR-$\tau$ curve for different attacks. It shows that although ASRs decrease with the increase of $\tau$, our attack still outperforms previous attacks.

| Victim model | | CosFace | MobFace | FaceNet |
|---|---|---|---|---|
| | Metric | ASR↑/FID↓ | ASR↑/LPIPS↓ | ASR↑/MSE↓ |
| Stage1 | w.o. Rec. loss | 38.25\79.65 | 39.85\69.26 | 44.46\88.43 |
| | w.o. LPIPS | 37.35\85.50 | 31.68\72.45 | 40.25\82.37 |
| | w.o. Dis. loss | 30.46\78.54 | 34.28\65.36 | 32.56\85.45 |
| Stage2 | w.o. Rec. loss | 38.34\81.45 | 39.64\68.35 | 43.35\80.53 |
| | w.o. LPIPS | 33.70\85.54 | 32.80\72.43 | 36.56\66.89 |
| | w.o. Dis. loss | 32.43\79.40 | 36.75\65.73 | 37.67\78.56 |
| | w.o. B.V. loss | 36.46\75.64 | 35.34\62.41 | 41.33\74.56 |
| | with all losses | 37.95\70.54 | 38.46\57.89 | 42.30\61.29 |

Table 3: Ablations on the loss functions. We compute the ASR (%), MSE, FID and LPIPS without different loss functions on the CelebA dataset.

**The influence of different loss functions** We systematically examine the impacts of different loss functions (Table 3). The findings revealed that the reconstruction loss yielded a noteworthy reduction in the MSE distance. However, it also harms the transferability. This may be because of the pixel-wise constraints restrict the perturbation space. The LPIPS loss can significantly decrease the perceptual distance with almost no harm to the ASR. Interestingly, we found that the boundary variance loss can significantly enhance the patch consistency while improving the transferability.

**Attack on defense models** To assess the ability to break defenses, we selected two robust models, PGD-AT (Madry et al., 2017) and TRADES (Zhang et al., 2019). Our attack models are trained by using ArcFace and FaceNet as white-box ensemble models. Most robust models employ adversarial training, making them highly resistant to gradient-based attacks like FGSM. However, we achieve ASRs of 52.3% and 35.5% for the PGD-AT and TRADES, as shown in Figure 7. We attribute the results to the fact that most robust models are trained on gradient-based examples and consequently cannot defend our attack, which manipulates the source image's deep features instead.

## 5 CONCLUSION

Previous adversarial patch attacks usually result in unnatural patterns, which are not stealthy. Their generated unnatural patterns also limit the adversarial patch's transferability. To solve these problems, this paper proposes a new attack called Adv-Inpainting to generate stealthy, inconspicuous, naturalistic, and high-transferable adversarial patches. Different from previous attacks, Adv-Inpainting has a two-stage coarse-to-fine framework that takes both the source and target images into consideration. The first stage generates transferable and natural-looking adversarial patches through attention-guided deep feature manipulation. The second stage further improves stealthiness by refining the patches. Experiments on various datasets and FR models demonstrate that our attack has better transferability and stealthiness than previous adversarial patch attacks and semantic attacks.

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

## A    ADDITIONAL RESULTS ON TRANSFERABILITY AND STEALTHINESS

We also do a cross-model evaluation on the transferability. We use Cosface as the white box model to extract identity embeddings and calculate the identity losses, and then evaluate the transferability on ArcFace, FaceNet, and MobFace. We compare our attack with the gradient-based, patch-based, and semantic-based attacks respectively. The ASR results are shown in Table 4. The results show that the transferability of our attack outperforms the previous gradient-based attacks, adversarial patch attacks, and semantic attacks by large margins.

| Dataset | LFW Dastaset | | | | CelebA Dataset | | | | FFHQ Dataset | | | |
|---|---|---|---|---|---|---|---|---|---|---|---|---|
| Victim Model | CosFace | ArcFace | MobFace | FaceNet | CosFace | ArcFace | MobFace | FaceNet | CosFace | ArcFace | MobFace | FaceNet |
| FGSM | 91.55 | 10.26 | 4.67 | 8.22 | 87.35 | 3.56 | 3.56 | 6.37 | 84.37 | 6.24 | 6.38 | 11.74 |
| PGD | 91.24 | 14.26 | 17.80 | 14.25 | 92.45 | 5.26 | 10.24 | 14.22 | 89.36 | 12.12 | 10.24 | 12.55 |
| C&W | 92.45 | 13.34 | 10.23 | 11.23 | 92.34 | 15.24 | 12.32 | 12.23 | 84.34 | 15.23 | 14.87 | 15.23 |
| MI-FGSM | 92.23 | 17.28 | 13.54 | 17.23 | 92.12 | 19.23 | 10.24 | 13.24 | 95.23 | 19.21 | 11.82 | 12.29 |
| Adv-Glasses | 52.12 | 5.23 | 4.54 | 7.25 | 26.32 | 9.21 | 12.56 | 8.92 | 56.34 | 9.23 | 7.21 | 11.31 |
| Adv-Hat | 23.64 | 12.43 | 19.21 | 14.62 | 72.34 | 12.45 | 10.23 | 7.36 | 63.23 | 11.83 | 14.32 | 12.63 |
| TIDIM | 82.12 | 26.33 | 28.32 | 21.27 | 82.18 | 22.27 | 19.23 | 20.25 | 17.28 | 19.21 | 27.32 | 22.57 |
| Gen-AP | 83.21 | 25.21 | 20.31 | 14.31 | 84.52 | 19.32 | 29.72 | 21.26 | 85.32 | 27.45 | 26.54 | 19.23 |
| SMAP | 82.45 | 21.35 | 21.76 | 28.98 | 82.81 | 30.12 | 29.43 | 28.65 | 91.21 | 30.13 | 32.56 | 32.35 |
| SemanticAdv | 92.79 | 10.89 | 14.27 | 6.32 | 91.23 | 24.67 | 21.47 | 10.22 | 82.43 | 16.23 | 17.42 | 10.88 |
| Adv-Makeup | 92.56 | 19.65 | 18.26 | 10.27 | 92.12 | 21.78 | 17.29 | 10.28 | 82.78 | 21.79 | 17.23 | 11.98 |
| Adv-Attribute | 82.71 | 26.54 | 18.27 | 12.78 | 89.27 | 14.29 | 18.20 | 28.37 | 82.39 | 27.32 | 29.21 | 28.45 |
| Ours w.o. AAIN | 92.54 | 32.54 | 27.56 | 31.26 | 88.32 | 28.46 | 29.42 | 32.96 | 87.39 | 29.40 | 29.20 | 31.68 |
| Ours w.o. APT | 94.32 | 36.37 | 31.28 | 38.32 | 89.39 | 37.62 | 33.29 | 39.26 | 93.21 | 32.86 | 33.24 | 39.67 |
| Adv-Inpainting | 94.29 | **41.27** | **38.29** | **41.26** | 94.28 | **34.29** | **39.57** | **42.38** | 92.18 | **38.72** | **27.26** | **41.28** |

Table 4: The attack success rates (%) of impersonation attacks for the different face recognition models on the LFW and CelebA datasets. We chose the CosFace model as the white-box model to train our models and evaluate the transferability of the other three FR models. We compare our attack with gradient-based, patch-based, and semantic-based attacks respectively.

We also compare our attacks with other attacks, as shown in Figure 8. The below numbers are the identity features' cosine similarity with the target images. The red numbers are on the white-box model ArcFace, and the blue numbers are on the black-box model FaceNet. Compared with previous adversarial patch attacks, although there still are some artifacts, our generated patch is more stealthy.

## B    ABLATIONS

We also evaluate the performance of different optimizers, as shown in Figure 9. The results show that AdamW slightly outperforms SGD and Adam. We do not use discriminator loss when evaluating the optimizers. It is shown that our training process is more stable compared to previous generative attacks. We think this is because the perceptual loss is not contradictory to the adversarial loss. Therefore, the training process is more stable.

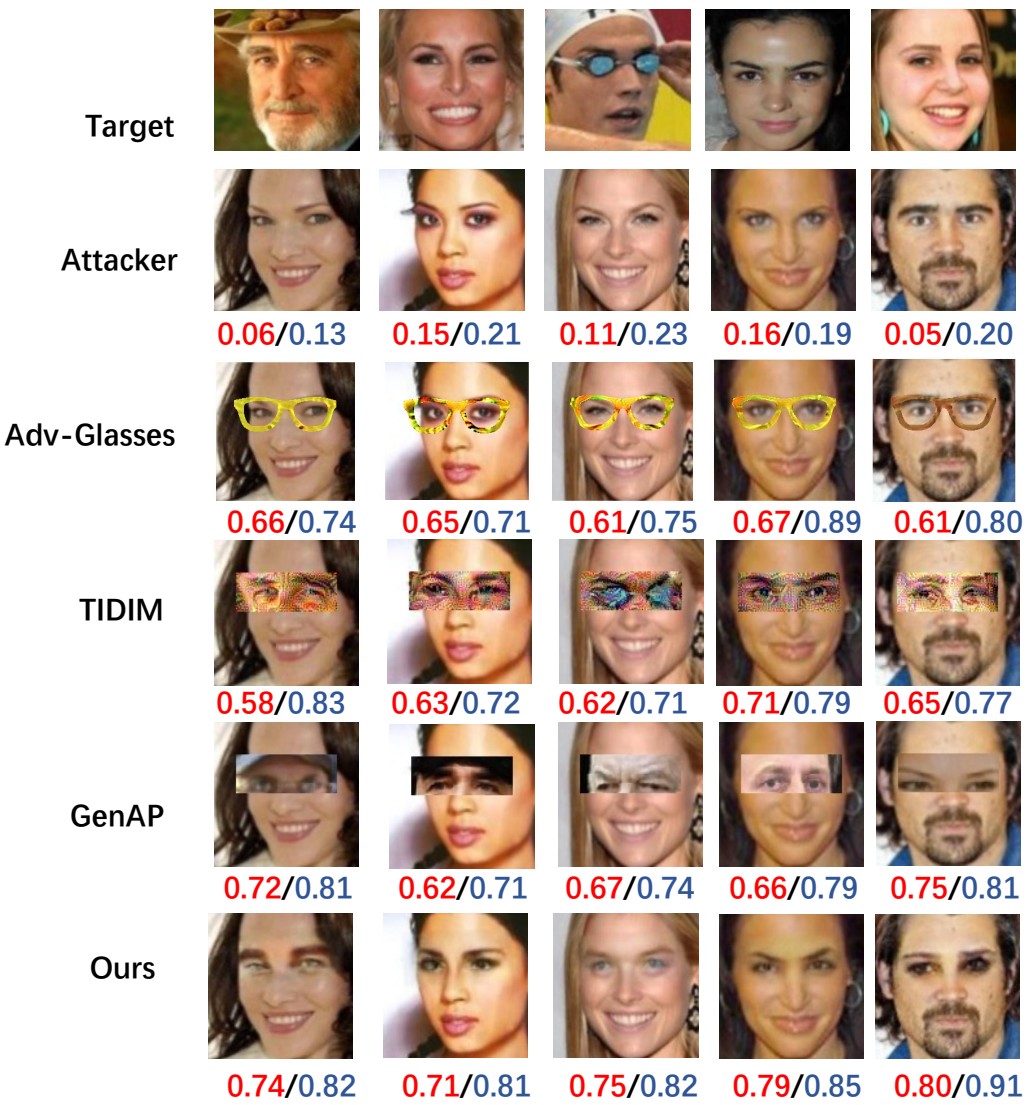

Figure 8: Visualization of different attacks. The below numbers are the identity features' cosine similarity with the target images. The red numbers are on the white-box model ArcFace, and the blue numbers are on the black-box model FaceNet.

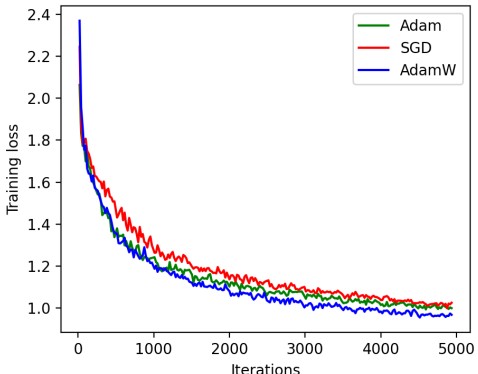

Figure 9: Impact of different optimizers.

