# OpenReview forum: "Generating Transferable and Stealthy Adversarial Patch via Attention-guided Adversarial Inpainting"
_ICLR.cc/2024/Conference — ICLR 2024 Conference Withdrawn Submission_

### Official Review · Reviewer_WMpX · 2023-10-27

**Soundness:** 3 good
**Presentation:** 3 good
**Contribution:** 2 fair
**Rating:** 5
**Confidence:** 3

**Summary:**

This paper proposes a two-stage method to generate transferable and stealthy adversarial patch. In the first stage, the embeddings of the style and identity are obtained by two encoders and be combined by cross-attention of the source feature and feature from last block. In the second stage, the authors propose an adversarial patch refinement network, which make adversarial patches more stealthy and coherent by the Boundary Variance Loss. Besides, LPIPS is used as a perception loss to keep patches similar to the original content. Evaluations on three datasets show its efftiveness.

**Strengths:**

1. The paper is well-organized and easy to understand.
2. The authors propose a new adaptive normalization based on the background-patch cross-attention map, which seems to be new and novel.
2. Empirical evaluations show a good improvement.

**Weaknesses:**

1. Formula (6) to calculate beta_i on Page 5 is wrong.
2. Formula (4) on Page 4 is not clear, and the Fig 2(b) is not consistent with this formula.
3. The whole framework contains two stages and a lot of modules, the time cost is important metric, which shouldn't be omitted.

**Questions:**

1. There are a lot of hyperparameters, is the method sensitive to these hyperparameters?
2. The authors choose to gain cross-attention map from F_in and F_p, can you give further explanation why it works and why other feature maps aren't chosen?
3. I'm not sure whether the idea of using attention in normalization is new. If so, I will increase my score.

---

### Official Review · Reviewer_ZKhL · 2023-10-29

**Soundness:** 3 good
**Presentation:** 3 good
**Contribution:** 3 good
**Rating:** 5
**Confidence:** 3

**Summary:**

In this paper, to generate transferable, natural-looking, and stealthy adversarial patches, the authors propose an innovative two-stage attack, which extracts style features and identity features from the attacker and target faces, respectively and then fills the patches with misleading and inconspicuous content guided by attention maps. Experiments demonstrate that their attack can generate adversarial patches with improved visual quality, better stealthiness, and stronger transferability than state-of-the-art adversarial patch attacks and semantic attacks.

**Strengths:**

- Propose a novel transferable, natural-looking, and stealthy adversarial patch attack
- Conduct extensive evaluation

**Weaknesses:**

- Lack of comprehensive evaluation on the natural-looking and stealthy

In this paper, the authors employ several metrics to evaluate the natural-looking and stealthy such as FID. However, this kind of metrics may not ensure the stealthiness and natural-looking. It will be beneficial if the authors can have a comprehensive evaluation through user studies. For instance, such evaluation is generally used in the existing works: the paper "Naturalistic Physical Adversarial Patch for Object Detectors" published in ICCV 2021. Such evaluation can improve their evaluation.

- Lack of the practicality

This paper is motivated by real-world application but later on there is no comprehensive evaluation on the practicality side. This paper also proposes a transferability-based black-box attack. Thus, by nature, they can evaluate their attack method on a commercial system. It would be great if the authors can provide such evaluation.

- Lack of novelty

Leveraging the attention maps to generate the adversarial patch attack is not novel such as "Attention-Guided Black-box Adversarial Attacks with Large-Scale Multiobjective Evolutionary Optimization" published in ICML 2021 workshop. It will be beneficial if the authors can provide detailed discussion on this works.

- Lack of comprehensive evaluation on defense side

The authors provide several discussion and evaluation on the defense. However, the evaluation is not very comprehensive and their evaluation target lacks justification on whether their selected targets are representative or not. Without such justification, it is unclear whether their attack can work under the most state-of-the-art defenses.

**Questions:**

Provide more discussion and evaluation on novelty, defense, practicality, and natural-looking.

---

### Official Review · Reviewer_iAwQ · 2023-11-01

**Soundness:** 2 fair
**Presentation:** 3 good
**Contribution:** 2 fair
**Rating:** 5
**Confidence:** 3

**Summary:**

This paper proposes a novel two-stage coarse-to-fine framework to generate adversarial patches. The proposed method aims to generate a transferable and stealthy patch which is based on the styleGAN combined with a refined stage to further improve the stealthiness of the adversarial patch. The experiment results show the proposed method can achieve state-of-the-art performance on the transferability and stealthiness.

**Strengths:**

1. The proposed method shows a good transferability and stealthiness of the adversarial patch.
2. The method analysis is clear and visualization is helpful.
3. The experiments are solid and comprehensive.

**Weaknesses:**

1. The statement about the patch selection is not clear.
2. The selection of the attacker and target is not clear.
3. Typo on Fig 5: 'MobileNet' -> 'MobFace'.

**Questions:**

1. According to the provided visualization, the adversarial patches are the attacker's whole face except the eyes part. Is this a fixed pattern? Can we use less part of the attacker?
2. When you training your model, are the attacker and target randomly picked or picked considering their cosine similarity from two different classes? If randomly picked, can we pick from most similar to least similar (training from easy to hard)?

---

### Official Review · Reviewer_uQV1 · 2023-11-06

**Soundness:** 3 good
**Presentation:** 2 fair
**Contribution:** 2 fair
**Rating:** 5
**Confidence:** 2

**Summary:**

This paper proposes a framework to generate adversarial patches using a two-stage attack called Adv-Inpainting. In the first stage, the method extracts multi-scale style embeddings and identity embeddings. In the second stage, the method includes an Adversarial Patch Refinement Network (APR-Net) with multiple loss terms including a boundary variance loss.

**Strengths:**

This paper works on generating transferable, natural-looking, and stealthy adversarial patches. It is an interesting topic and worth investigating.

**Weaknesses:**

The writing of the paper needs significant improvements. Even some of the formulations are not correct. The annotations are quite confusing and make it hard to read. For instance, the $\beta_i$ in Eq. (6) is not correct. Eq. (10) also looks to be not correct, the $h$ and $w$ are supposed to be the footnote or something.

It is also not clear how the mask is chosen for each image. The location of the patch is also a very important aspect that highly influence the performance.

**Questions:**

1. How is the selection of the white-box model influence the final performance? In the paper, the authors choose ArcFace model as the white-box model. What is the performance by setting the white-box model as CosFace, MobFace, and FaceNet, respectively, and testing on the other three model as black-box model cases?

2. Does the good performance come from the similarities among different face recognition models? Maybe different FR extracts similar feature maps, making the method trained on one white-box model transferable to other FR models. Is the attack success rate correlated with the model similarity?

3. How is the mask selected for different images? Is it fixed for all the images or customized for each image?